# Factors Associated with Quality of Life of Clinical Nurses: A Cross-Sectional Survey

**DOI:** 10.3390/ijerph20031752

**Published:** 2023-01-18

**Authors:** Seul-Ki Park, Kyoung-Sook Lee

**Affiliations:** 1Ulsan University Hospital, 25 Daehakbyeongwon-ro, Dong-gu, Ulsan 44033, Republic of Korea; 2Department of Nursing, University of Ulsan, 93 Daehak-ro, Nam-gu, Ulsan 44610, Republic of Korea

**Keywords:** quality of life, nurses, stress, sleep quality

## Abstract

Nurses often have a heavy workload and struggle to maintain a good quality of life (QOL). The aim of this study was to explore the relationship between perceived stresses and sleep disturbance, and quality of life among Korean clinical nurses. A cross-sectional design was used to examine these relationships among 200 clinical nurses at three hospitals in South Korea. Standardized instruments were used, including the Perceived Stress Scale (PSS), Insomnia Severity Index (ISI), Eating Behavior Questionnaire (DEBQ), and the WHOQoL-BREF questionnaire. The data were analyzed using descriptive statistics, the t-test, ANOVA, the Scheffe test, Pearson’s correlation coefficient, and multiple regression analysis using SPSS/WIN 24.0 program. Multiple regression analysis showed that clinical nurses who had a subjective health status (*β* = 0.29, *p* = 0.001), perceived stress (*β* = −0.32, *p* < 0.001), and sleep disturbance (*β* = −0.21, *p* = 0.001) were more likely to have a higher quality of life. The explanatory power of the regression model was statistically significant at 36.7%. Multiple regression analysis showed that clinical nurses who had better subjective health status, lower perceived stress, and fewer sleep disturbances were more likely to have a higher quality of life.

## 1. Introduction

The hospital environment and long time with patients cause an increase in the physical and psychological burden of nurses and a decrease in the quality of life [1]. In Korea, only half of the licensed nurses serve as clinical nurses. Among the OECD members, the country has the lowest ratio of licensed to clinical nurses [2]. Most nurses deal with varying work shifts, night shifts that decrease the quality of life, staff shortages that increase workload and work intensity [3], a high complaint rate from patients and guardians that induce stress [4], and a hierarchical and authoritative hospital culture [5]. For these reasons, nurses have a short continuous service period. The number of nurses on shift work who took sleeping pills or drank alcohol due to sleep disturbance was 3.3 times higher than that of people with other occupations, and the rate of nurses claiming “it is difficult to build social relationship due to work” was 3.1 times higher [5]. Quality of life is one’s subjective well-being status in physical, psychological, and social areas [6]. When nurses’ quality of life increases, their professionalism improves, thus enabling them to provide higher-quality nursing services to patients [7]. However, when the quality of life diminishes, they are unable to pay proper attention to patients, which may negatively influence patient care [8]. Therefore, nurses’ quality of life is crucial, and the factors affecting it must be elucidated.

In their line of work, nurses are exposed to health-threatening risk factors, such as biologically hazardous and carcinogenic substances, infectious diseases, and shift work [7,8]. Nurses are inevitably under a lot of stress, which can affect their health negatively [9,10,11,12,13]. Compared with employees in other professional areas, nurses have higher perceived stress [14]. If their stress is repeated, persists, or is increased, it negatively influences them physically and psychologically. Since stress not only reduces nurses’ quality of life but also negatively affects their professional roles, including providing patient care [15], helping nurses manage perceived stress is very important.

Nurses’ rotating work shifts disrupt their biorhythms and cause a continuously irregular life pattern, which may lead to sleep disorders, such as insomnia, and poor sleep quality [16]. Sleep disturbance not only decreases nurses’ work performance, but also increases the risk of work-related mistakes, such as inaccurate interpretation of patient conditions, incorrect operation of medical devices, and needle-induced injuries [17]. Moreover, nurses’ continuous sleep disturbance negatively affects their physical, psychological, and mental well-being [16], causing chronic diseases [18], and lowering work efficiency [19]. Consequently, sleep disturbances can decrease the quality of life of nurses on rotating work shifts.

Nurses carry the duty of providing professional nursing services to patients and of keeping good relationships with different teams in the workplace, placing them under more stress than employees in other professional areas [20]. Some ways of coping with stress include overeating, which can erode one’s self-confidence in dietary control [20]. According to a study by Szweda and Thorne [21], 20% of nurses exhibit abnormal eating behavior, whereas McNutly [22] reported 49.6%. Korean nurses on shift work take irregular meal times and develop unhealthy dietary behaviors overall [23], such as eating snacks at night and being underweight [24]. For this reason, nurses are highly susceptible to adopting abnormal eating behavior, which can negatively influence their quality of life. Therefore, nurses’ eating behaviors are a significant subject of study.

Generally, the work of nurses is evaluated to be hard, given their work conditions, subjective health status, job satisfaction, shift work, work-related fatigue, living habits, and psychological status. Much research has been conducted on improving nurses’ quality of life. Since these correlations have been verified, the necessity to improve the quality of life of nurses as primary caregivers has been raised [25,26].

A previous study investigating conditions where the quality of life was set as a dependent variable found a correlation between nurses’ quality of life and work factors, including work stress. Regarding the quality of life of nurses on shift work, fewer sleep disturbances [27,28] resulted in more positive health behaviors and, hence, a better quality of life [27]. However, the analyses on the relationship among nurses’ perceived stress, sleep disturbance, and eating behavior were limited. In-depth research on these factors needs to be conducted to improve nurses’ quality of life.

The purpose of this study was to explore the relationship between perceived stress sleep disturbance and eating behavior, and the quality of life among Korean clinical nurses.

## 2. Methods

### 2.1. Study Design

This work adopted a cross-sectional study design to determine how perceived stress, sleep disturbance, and eating behavior affect nurses’ quality of life (Figure 1).

### 2.2. Study Participants

In this study, convenience sampling was performed to recruit nurses at three general hospitals in Ucity. To determine the number of samples, the G*Power 3.1 program was used for analysis and verified using the parameters needed for regression analysis: a significance level of 0.05, explanatory power of 0.95, medium effect size of 0.15, and 13 predictor variables. The number of samples determined was 189. After considering a dropout rate of 10%, a total of 210 clinical nurses were enrolled as study participants. The detailed acceptance criteria of the hospitals and participants are as follows: to be included in the study, a hospital must (1) be a second or tertiary hospital (referral center for primary and secondary levels of care) (2) have at least 150-bed capacity (3) have at least 350 nurses. (4) be the one where the head of the nursing department understood the purpose of the study and allowed the completion of the questionnaire. Participants were included from any of the three selected hospitals. They were fluent in the Korean language, able and willing to complete the questionnaire, and granted informed consent.

#### Study Population

There was one tertiary hospital and two secondary hospitals that met the criteria. About 1400 nurses worked in the tertiary hospital, and two secondary hospitals were hospitals with about 350 nurses. The questionnaire was distributed to 10% of the nurses working at each hospital. Each hospital was required to fill out a questionnaire in consideration of the department of work and the presence or absence of shift work.

### 2.3. Measures

#### 2.3.1. Variables

The general characteristics used in this study were a total of 10 questions, including age, marital status, education level, family monthly income, subjective health status, working department, career experience, shift work, night shift, and job satisfaction, and perceived stress, sleep disorders, dietary behavior, and quality of life.

#### 2.3.2. Perceived Stress Scale (PSS)

The PSS, which was developed by Cohen et al. [27] and modified by Park and Seoh [28], was used. It is based on a five-point scale ranging from “Strongly disagree (0 points)” to “Strongly agree (4 points)” and consists of 10 questions. In the PSS, the higher the total score, the higher the perceived stress level. The subfactors of perceived stress are positive and negative perceived stress. In the study by Cohen et al. [27], the Cronbach’s ⍺ of the PSS was 0.87. In the study by Park and Seoh [28], the Cronbach’s ⍺ of the scale was 0.85, whereas that of the subfactors was not revealed. In the present study, the Cronbach’s ⍺ of the scale was 0.78, whereas that of positive and negative perceived stress was 0.77 and 0.86, respectively.

#### 2.3.3. Sleep Disturbance (Insomnia Severity Index, ISI)

To measure sleep disturbance, the ISI developed by Morin [19] and translated by Cho [29] was used. It consists of seven questions and is scored using a five-point Likert scale (0–28 points). The ISI has been found suitable for measuring the sleep disturbance of shift workers [17]. In this study, the ISI was used to measure the latest 2 week insomnia and sleep quality. The higher the total score, the greater the sleep disturbance. At the time when the original instrument [30] was developed, Cronbach’s ⍺ = 0.74. The Korean version ISI [29] had a Cronbach’s ⍺ = 0.92. In this study, the Cronbach’s ⍺ of the instrument was 0.91, whereas that of insomnia and of sleep quality was 0.89 and 0.82, respectively.

#### 2.3.4. Eating Behavior

The Dutch Eating Behavior Questionnaire (DEBQ) was developed by van Strien et al. [31] and translated by Kim et al. [32]. It was developed to evaluate the eating habits of people with normal weight and overweight. The translated version of the instrument was used after the translator’s approval. DEBQ has three subfactors: restrained eating, emotional eating, and external eating comprising 33 questions that are scored using the five-point scale. The higher the score, the more abnormal the eating behavior. When the DEBQ was developed [31], the Cronbach’s ⍺ of restrained eating was 0.95, that of emotional eating 0.91, and that of external eating 0.80. In the study by Kim et al. [32], the Cronbach’s ⍺ of restrained eating was 0.90, that of emotional eating 0.93, and that of external eating 0.79. In this study, the Cronbach’s ⍺ of the instrument was 0.93, that of restrained eating 0.94, that of emotional eating 0.96, and that of external eating 0.87.

#### 2.3.5. Quality of Life

To assess the quality of life, the World Health Organization (WHO) developed WHOQOL-BREF [33]. Min et al. [34] modified the WHOQOL-BREF and developed the WHOQOL-BREF: Korean version. In this study, the Korean version was used, which consisted of 26 questions on five domains: physical health, psychological health, social relationships, environment, and overall quality of life. Each question is scored based on the 5-point Likert scale, with answers ranging from “Strongly disagree (1 point)” to “Strongly agree (5 points)”. The higher the score, the higher the quality of life. At the time when the WHO developed the questionnaire [33], Cronbach’s ⍺ of physical health was 0.84, that of psychological health was 0.76, that of social relationships was 0.66, and that of environment was 0.80. The overall quality of life domain was not assessed. In the Korean version [34], the Cronbach’s ⍺ of the instrument was 0.90, that of physical health was 0.78, that of psychological health was 0.76, that of social relationships was 0.58, that of environment was 0.77, and that of overall quality of life was 0.59. In this study, the Cronbach’s ⍺ of the instrument was 0.93, that of physical health was 0.80, that of psychological health was 0.81, that of social relationships was 0.73, that of environment was 0.81, and that of overall quality of life was 0.70.

### 2.4. Data Collection

This study was conducted after approval from the institutional review board of hospital A of the research design and method (IRB No.: UUH202007019002-HE001). The participants were nurses and head nurses at three general hospitals with more than 150 beds each in U metropolitan city and received cooperation and approval in accordance with an official procedure. Data were collected through a questionnaire survey from 16 September to 15 October 2020. Before filling in the questionnaire, the study participants received an explanation of the study’s purposes, the data collection method, and the strict use of the data collected for only the purposes stated. The researcher visited the selected hospital ward and distributed the questionnaire to the shift workers. The completed questionnaire was sealed and placed in a container after one week. When the questionnaires were filled, the researcher went and collected them herself. The questionnaire was placed at the nurse’s office desk, and the desired nurse took it voluntarily to respond to the questionnaire and did not include information that could identify the study participants through the data. The collected data were prepared and enclosed in an envelope, and a collection box was placed in front of the restaurant to be collected by the researcher every three days. The data were organized and analyzed by the researcher and he did not try to backtrack or identify the participants. The consent form and survey data will be kept for three years and then discarded, and the coded data will be kept for at least five years before being deleted. A total of 210 questionnaire copies were distributed to those who agreed to participate in this study, and 201 copies were collected. However, one with insincere answers was excluded. A final total of 200 responses were analyzed.

### 2.5. Data Analysis

In this study, SPSS Win 24.0 was used to conduct the data analysis. The details of the analysis are as follows:(1)The general characteristics of the study participants were expressed as mean and standard deviation or frequency and percentage.(2)The study participants’ perceived stress, sleep disturbance, eating behavior, and quality of life were expressed as mean and standard deviation.(3)The study participants’ quality of life was analyzed according to their general characteristics using the independent t-test and one-way ANOVA after conducting the normality test. The Scheffe test was conducted as a Post-Hoc test.(4)The correlations among the study participants’ perceived stress, sleep disturbance, eating behavior, and quality of life were analyzed using Pearson’s correlation coefficient.(5)To identify the factors influencing the study participants’ quality of life, multiple regression analysis was performed. All tests were the significance considered as *p* < 0.05.

### 2.6. Bias

The sample size was determined using the G power 3.1 programs. When recruiting the participants, convenience sampling was performed at each hospital, and 10% of the nurses were selected from the three hospitals. Moreover, efforts were made to meet the ratio in consideration of each department, age, and shift work. By verifying the normality for regression analysis, the participants could assume a normal distribution for each variable, and only one of all the collected questionnaires had a few missing values.

## 3. Results

### 3.1. Quality of Life Based on General Characteristics

Table 1 shows that the participant’s quality of life varied significantly depending on their subjective health status (t = 6.03, *p* < 0.001) and job satisfaction (F = 9.45, *p < 0*.001). For the subjective health status, nurses who replied “moderate or good” had a higher quality of life than those who replied “poor”. There was a difference in the quality of life according to job satisfaction, and as a result of post-hoc analysis (Scheffe), nurses who replied “Satisfaction” had a higher quality of life than those who replied “moderate”. Nurses who replied “moderate” had a higher quality of life than those who replied “dissatisfaction”. Detailed characteristics are shown in Table 1.

Table 1 shows that a total of 200 nurse participants were included in this study. Their average age was 31.55 ± 7.39 years. Regarding marital status, 123 participants (61.5%) were unmarried. Regarding education level, 111 participants (55.5%) graduated from university. Regarding monthly family income, 82 participants (41.0%) earned over 4166 dollars. In terms of subjective health status, 118 participants (59.0%) replied “moderate or good”. Concerning the work department, 60 participants (30.0%) worked in the medical unit, 54 participants (27.0%) in the surgical unit, 51 participants (25.5%) in the outpatient department, and 35 participants (17.5%) in a special unit (emergency room, intensive care unit, etc.). Regarding career experience, the average career length mean (SD) was 8.70 (7.11) years, and 74 participants (37%) had over 10 years of experience. With regard to shift type, 139 participants (69.5%) were on shift work; 61 participants (30.5%) were not on shift work; 127 participants (63.5%) worked the night shift; and 61 participants (30.5%) did not work the night shift. Regarding job satisfaction, 114 participants (57.0%) replied “moderate satisfied”, 44 participants (22.0%) replied “satisfied”, and 42 participants (21.0%) replied “dissatisfied.” (Table 1).

### 3.2. Study Subjects’ Perceived Stress, Sleep Disturbance, Eating Behavior, and Quality of Life

The results of the analysis of the subjects’ quality of life are presented in Table 2. The participants’ average quality of life score mean (SD) was 3.30 (0.46/5). For the sub-domains, the average score for the environment domain was 3.41 (0.52); for the social relationship domain, it was 3.35 (0.54); for the physical health domain, it was 3.28 (0.56); for the psychological domain it was 3.27 (0.53); and for the overall quality of life domain, it was 3.06 (0.62). Detailed characteristics are shown in Table 2.

### 3.3. Correlations between Variables

The correlations among the subjects’ perceived stress, sleep disturbance, eating behavior, and quality of life are presented in Table 3.

Perceived stress was positively correlated with sleep disturbance (r = 0.43, *p* < 0.001) and eating behavior (r = 0.30, *p* < 0.001) but significantly negatively correlated with quality of life (r= −0.53, *p* < 0.001). Sleep disturbance was positively correlated with eating behavior (r = 0.20, *p* = 0.004) but negatively correlated with quality of life (r= −0.43, *p* < 0.001). Eating behavior was negatively correlated with quality of life (r= −0.23, *p* < 0.001).

### 3.4. Factors Influencing Quality of Life

To identify the factors influencing clinical nurses’ quality of life, a multiple regression analysis was conducted. The results are presented in Table 4.

Multiple regression analysis was analyzed for general characteristics including age, marital status, education level, family monthly income, subjective health status, working department, career experience, shift work, night shift, job satisfaction, perceived stress, sleep disorders, and eating behavior. The assumption of the normal distribution of the sample was verified with absolute values of skewness and kurtosis. As a result of checking the Z-value of univariate normality in this study, none of the measured variables exceeded the absolute value of skewness 3, the absolute value of kurtosis 7, and the absolute value of Z-value, so it was judged that there was no particular problem in the sample normality review.

Among the predictor variables, the variable measured on a nominal scale was defined as the dummy variable. To test residual independence, the Durbin–Watson test statistic was calculated, which was 1.99, and no auto-correlation was observed for error terms. Therefore, residual independence was met. The tolerance of each variable was 0.43–0.89, indicating that no value was lower than 0.10. The variance inflation factor of each variable was 1.12–2.31, which was less than 10; thus, the variables had no multicollinearity.

The regression model of clinical nurses’ quality of life was significant (F = 39.38, *p* < 0.001). The regression model analysis showed that quality of life is affected by subjective health status (*β* = 0.29, *p* = 0.001), perceived stress (*β*= −0.32, *p* < 0.001), and sleep disturbance (*β*= −0.21, *p* = 0.001). The regression model of these variables had an explanatory power of 36.7%. In this study, age, marital status, education level, family monthly income, working department, career experience, shift work, night work, and job satisfaction did not significantly affect the quality of life.

## 4. Discussion

The present study identified the factors influencing nurses’ quality of life to be subjective health status, perceived stress, and sleep disturbance.

Among the general characteristics of this study, only subjective health status affected the quality of life and was the major factor influencing the quality of life. This is consistent with the results of a study of nurses in small and medium-sized hospitals that showed that the better the overall health condition, the higher the quality of life [3]. Moreover, it is similar to, the quality that there was no significant difference between age, income, medical center, education level, workplace, clinical experience, marriage, drinking and night work, and marital status, in a study of shift work nurses in small and medium-sized hospitals or a general hospital [3,34]. However, a study reported that age, relocation, economical facts, and education had a significant effect on the quality of life [3,35,36]. Therefore, a nurse’s subjective health status is influenced by healthful physical activities and living habits, and a perception of health is important to nurses [34]. Since subjective health status was identified as an influential factor in nurses’ quality of life, nurses need to have a positive subjective perception of health status and motivation to maintain their physical and psychological health. In addition, programs for improving nurses’ physical and psychological health need to be developed in medical organizations to establish active support.

The present study also demonstrates perceived stress as an influence on the quality of life. This finding agrees with a study that found that nurses’ perceived stress negatively affected their happiness [37]. A study involving nursing students [38] reported that lower perceived stress indicated higher quality of life. This information is also supported by the findings in the present study. These findings suggest the necessity of devising a plan to reduce nurses’ perceived stress. Although nursing is a stressful profession, the aspects related to nurses’ health are often overlooked [37]. Hence, it is necessary to apply various step-by-step and integrated stress management programs for nurses.

Finally, this study found that sleep disturbance was an influential factor in the quality of life. The participants of the present study had more than a moderate level of sleep disturbance on average. This is similar to the result that 57.8% of new nurses and 67.6% of career nurses had poor sleep quality in Yu’s study of shift nurses [39]. Good sleep is very important to nurses who are caring for patients [40]. Nurses experience a lot of daytime dysfunction due to lack of sleep, which negatively affects other health conditions or quality of life [41]. Because shift work is a major factor that can cause sleep disturbances and deteriorate sleep quality [9] when deciding on work in shifts, it is desirable to consider the health status, career, and age of nurses, and more active intervention should be made to improve sleep according to individual sleep types [41]. The ways to improve nurses’ sleep quality can include aroma therapy [42], which was found to be effective against sleep disturbance, and insomnia cognitive-behavior therapy program, which is based on mobile social networking services [42]. Henceforth, in order to improve the clinical nurses’ quality of life, interventions to promote sleep suitable for individual sleep types should be more actively performed.

In this study, we observed that eating behavior did not significantly affect the quality of life. However, the correlation between eating behavior and quality of life was significant, which means that if there was a problem with eating behavior is related to poor quality of life. The eating behavior of the participants of this study was low at 2.82 points out of 5. It was found that 51.8% of nurses on shift had normal eating behavior and 44% had poor eating habits. In particular, it is similar to a study where no nurse answered “excellent” for eating habits [43]. As a result, measures such as increasing the operating time of the hospital restaurant and operating a well-nourished specialty should be improved so that people can eat even if they miss time at the hospital restaurant. In addition, in order to ensure meal time during work, it is necessary to enable efficient operation of work and to provide additional support for nursing personnel by introducing a flexible work system at high hours of work.

Quality of life is an indicator that best reflects an individual’s well-being by individual subjective judgment rather than the influence of an objective environment, and in recent studies, studies using quality of life as a result variable are gradually increasing [36]. It is necessary to carry out such studies by adding various variables for the quality of life. Furthermore, various policy decisions and personal efforts will be required to improve the quality of life of nurses.

This study identified factors that can share the most time with patients and improve the quality of life of nurses who affect them. Since the quality of life of nurses is directly related to the quality of patient care, improving the quality of life of nurses will lead to improving the quality of patient care and the image of medical institutions. In terms of nursing practice, through the results of this study, it was possible to identify the subjective health status of nurses, perceived stress, sleep disorders, the status of quality of life, and related factors and causal relationships.

## 5. Conclusions

To conclude, it was confirmed that clinical nurses who had a better subjective health status, lower perceived stress, and fewer sleep disturbances were more likely to have a higher quality of life. Therefore, plans should be established to improve the quality of life by improving the subjective health status of clinical nurses, reducing perceived stress, and minimizing sleep disorders.

## 6. Limitations

This study has some limitations. First, convenient sampling was applied in enrolling clinical nurses from three general hospitals in one city. Hence, caution should be exercised in generalizing the study results. Second, the number of male nurses is increasing; however, all the study participants were female, which limits the generalizability of the findings to male nurses. Third, errors inherent to a self-reported questionnaire could not be excluded. Specifically, the accuracy of the assessment for eating behavior could not be guaranteed since it was based on the subjects’ personal judgment, and excluding any errors in filling in the questionnaire was impossible. Fourth, in this study, alcohol, smoking, and physical activity that can affect the quality of life were not included.

Based on these limitations, the following measures for obtaining more decisive results are suggested:

First, since convenient sampling was applied, and there is limited the generalizability of results, further research involving larger samples in diverse nursing settings needs to be conducted. Moreover, it is suggested to expand the research and study the participants by including alcohol drinking, smoking, and physical activity variables that were not included in this study. Second, given that the participants are clinical nurses, further in-depth research categorizing according to work type could help broaden the implications. Third, since this study is bound by the limits of a questionnaire-based survey, more qualitative study methods that include the evaluation of BMI, various measurement instruments, interviews, and participant observation will need to be employed in further studies. Fourth, the results of this study highlight the importance of developing and applying an integrated education program to improve nurses’ quality of life.

## Figures and Tables

**Figure 1 ijerph-20-01752-f001:**
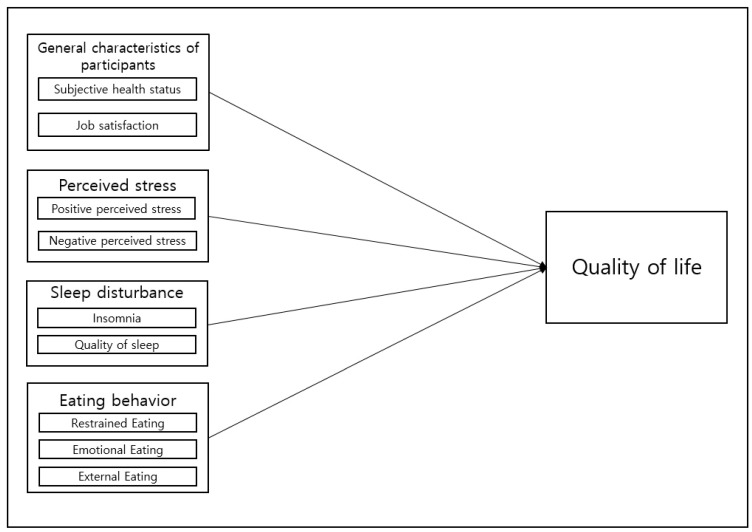
Conceptual framework of the study design.

**Table 1 ijerph-20-01752-t001:** Quality of life based on general characteristics (*N* = 200).

Characteristics	Category	n	%	Mean (SD)	Quality of Life	
					Mean (SD)	t/F(*p*) Scheffe
Age (years)	≤25	43	21.5	31.55 (7.39)	3.37 (0.39)	0.66 (0.622)
	26–30	74	37.0		3.24 (0.44)	
	31–35	23	11.5		3.34 (0.62)	
	36–40	32	16.0		3.27 (0.45)	
	≥41	28	14.0		3.32 (0.47)	
Gender	Female	200	100		3.30 (0.46)	-
	Male	0	0		-	
Marital status	Unmarried	123	61.5		3.28 (0.45)	0.44 (0.662)
	Married	77	38.5		3.31 (0.48)	
Education level	Diploma	57	28.5		3.29 (0.43)	1.50 (0.226)
	Bachelor	111	55.5		3.26 (0.46)	
	≥Master	32	16.0		3.42 (0.53)	
Family income (doller /month)	<2500	59	29.5		3.19 (0.46)	2.04 (0.133)
	2500–4158	59	29.5		3.33 (0.48)	
	≥4166	82	41.0		3.35 (0.44)	
Subjective health status	Poor	82	41.0		3.08 (0.45)	6.03 (<0.001)
	Moderate or good	118	59.0		3.44 (0.40)	
Work department	Medical unit	60	30.0		3.27 (0.46)	0.13 (0.943)
	Surgical unit	54	27.0		3.32 (0.46)	
	Special unit	35	17.5		3.32 (0.42)	
	OPD	51	25.5		3.29 (0.50)	
Career experience (years)	<1	17	8.5	8.70 (7.11)	3.46 (0.40)	0.74 (0.563)
	1–3	30	15.0		3.27 (0.46)	
	3–5	27	13.5		3.24 (0.43)	
	5–10	52	26.0		3.27 (0.43)	
					3.31 (0.51)	
	≥10	74	37.0		3.32 (0.51)	
Shift work	Yes	139	69.5		3.28 (0.44)	0.51 (0.608)
	No	61	30.5		3.32 (0.51)	
Night shift	Yes	127	63.5		3.29 (0.44)	0.12 (0.903)
	No	73	36.5		3.30 (0.49)	
Job satisfaction	Dissatisfaction ^a^	42	21.0		3.02 (0.49)	9.45 (<0.001) a < b < c
	Moderate ^b^	114	57.0		3.29 (0.41)	
	Satisfaction ^c^	44	22.0		3.56 (0.42)	

Special units: emergency room, intensive care unit, OPD: outpatient department

**Table 2 ijerph-20-01752-t002:** Level of perceived stress, sleep disturbance, eating behavior, and quality of life (*N* = 200).

Variable Subdimensions	Mean (SD)	Range	Min.	Max.
Perceived stress	1.50 (0.83)	0–4	0	4
Positive perceived stress	1.90 (0.49)		0.2	3.4
Negative perceived stress	1.94 (0.70)		0	3.4
Sleep disturbance	1.92 (0.46)	0–4	0.6	3.1
Insomnia	1.40 (0.98)		0	4
Sleep quality	1.57 (0.82)		0	4
Eating behavior	2.82 (0.61)	1–5	1.24	4.61
Restrained eating	2.78 (0.89)		1	4.8
Emotional eating	2.48 (1.01)		1	5
External eating	3.30 (0.68)		1.4	5
Quality of life	3.30 (0.46)	1–5	2.12	4.69
Physical health domain	3.28 (0.56)		1.86	5
Psychological domain	3.27 (0.53)		1.83	4.83
Social relationships domain	3.35 (0.54)		2	5
Environmental domain	3.41 (0.52)		2	5
Overall quality of life and general health	3.06 (0.62)		2	5

**Table 3 ijerph-20-01752-t003:** Correlations among perceived stress, sleep disturbance, eating behavior, and quality of life (*N* = 200).

Variables	Perceived Stress	Sleep Disturbance	Eating Behavior	Quality of Life
	r (*p*)	r (*p*)	r (*p*)	r (*p*)
Perceived stress	1			
Sleep disturbance	0.43 (<0.001)	1		

Eating behavior	0.30 (<0.001)	0.20 (0.004)	1	

Quality of life	−0.53 (<0.001)	−0.43 (<0.001)	−0.23 (0.001)	1

**Table 4 ijerph-20-01752-t004:** Factors influencing quality of life.

Variables	Quality of Life				
	B	S.E.	Β	t	*p*-Value
(Constant)	3.41	0.22		15.45	<0.001
Age	−0.52	−0.89	−0.23	−1.28	0.203
Marital status	−0.02	−0.09	−0.02	−0.26	0.793
Education level	−0.01	0.06	−0.10	−0.16	0.871
Family income	0.01	0.02	0.02	0.36	0.722
Subjective health status	0.37	0.08	0.29	4.49	<0.001
Work department	−0.01	0.03	0.02	−0.19	0.849
Career experience	−0.10	0.01	1.39	0.80	0.425
Shift work	−0.05	0.87	1.32	−0.74	0.460
Night shift	−0.03	0.06	0.23	−0.48	0.632
Job satisfaction	0.27	0.45	0.24	0.59	0.550
Perceived stress	−0.48	0.08	−0.32	−4.67	<0.001
Sleep disturbance	−0.15	0.48	−0.21	−3.32	0.001
Eating behavior	−0.05	0.05	−0.06	−1.01	0.321

R^2^ = 0.376, adjusted R^2^ = 0.367, F = 39.38, *p* < 0.001, Durbin–Watson = 1.99.

## Data Availability

Data supporting reported results can be requested from the corresponding author (favorseulki@uuh.ulsan.kr).

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
