# Peer review of "Factors Associated with Quality of Life of Clinical Nurses: A Cross-Sectional Survey"

_ijerph, 2023, doi:10.3390/ijerph20031752_

Round 1

Reviewer 1 Report

Dear Authors

I have read your MS entitled "The Influence of Clinical Nurses’ Perceived Stress, Sleep 2 Disturbance, and Eating Behavior on Quality of Life" with great interest.  Nurses QoL is influenced my many factors including lifestyle factors and life satisfcation.

Below are my comments: 

The MS needs editing and revision.

For example, I found the following errors:

1-      last paragraph of introduction

However, he analyses on the relationship among nurses’ perceived stress, sleep disturbance, and eating behavior were limited.

Correct: he to the

2-      Results

Line 12: 127participants (63.5%) 190 worked the night shift; and 61 participants (30.5%) did not work the night shift

However, in table 1, 30.5% did not do shift work.  The authors wrote: 61 participants (30.5%) did not work the night shift

3-      Authors should refer which table in the results they are explaining. For example, table 1 shows that “The participants’ quality of life varied significantly depending on their subjective 195 health status (F=23.17, p<.001) and job satisfaction (F=17.23, p><.001)……etc

4-       The authors wrote: “In general, higher the job satisfaction, better the quality of life”.  Edit the sentence

5-      Under the table please add the abbreviations meanings!!!  

What factors were included in the model?  What confounders are controlled for in the regression model?

The authors did not add in the model the effect of external factors like life satisfaction which is directly affects the QoL. Also, other lifestyle factors such as physical activity, smoking, alcohol is very important factors that determine QoL of nurses

What are the practical implications of the study? What are the policy implications of this study?

Author Response

I attached it as a file.

Reviewer 2 Report

Title

1.       The title can be revised to show that the major analysis was regression/correlation coefficient. The term “the influence of…” is exaggerated bearing in mind the simplistic nature of the data analysis.

2.       Add the study type to the title.

Abstract

1.       Please revise the writing style to present your points in a scholarly way. E.g., Line 8. “…the nurse lives a happy and human life.”

2.       If word count permits, state the study design, sampling technique, mode of questionnaire administration, and the names of the questionnaires. The date of data collection is not necessary for the abstract.

3.       Start the result section with brief descriptive statistics especially the age and gender of participants. Word the result section in one direction, preferably stating factors and QOL.

4.       Lines 20 to 23 should be revised to show a correlation/relationship rather than causal effects. Good QOL is far beyond good sleep, less stress, and healthy eating as presented here and throughout the manuscript.

Keywords

Maximize indexing by using MeSH keywords not already in the title.

Introduction

1.       Well written to some extent. A reasonable amount of information was captured in the introduction, but things needed to be rearranged and tidied up to improve the flow of text. The introduction needs English language copy editing. There are some grammatical errors E.g., in Line 80.

2.       The first paragraph is wordy and contains a lot of repetition of concepts or terms.

3.       The conceptual framework (Figure 1) can be better integrated into the introduction section.

4.       Lines 82 to 85 were overstated and in some cases beyond the scope of the present manuscript.

5.       Please end the introduction section with specific research questions or hypotheses that were treated in the data analysis section.

Methodology

1.       The study will benefit from being revised with the STROBE guidelines. Make the inclusion and exclusion criteria for (1) hospitals and (2) participants clear (see https://doi.org/10.1186/s12913-022-08808-3).

2.       Please revise the “study design.” It appears to be a cross-sectional survey.

3.       Add a subheading “Study population or context” to explain where this study was conducted and the nature of the study population. What are the population estimates of nurses in each of the three hospitals used? What is the population proportion of your final sample size relative to each hospital? Did you apply quota sampling apart from the convenience sampling you mentioned?

4.       Add a subheading “Variables” to explain the nature and scale of measurement of all the primary outcomes and the predictors/covariate including the sociodemographic data.

5.       Add the subheading “Bias” to discuss how the recruitment/sampling technique and sample size may have or have not affected the representativeness of the final sample.

6.       Expand the data collection procedure a little further. Was it an online or hard copy questionnaire, administered by hand or mailed? How were the nurses approached, during shift duty or in a workshop? Was the questionnaire completed and collected immediately or did the researchers return to collect them, or they were mailed to the researchers? The explanation should be enough to ensure the replicability of the study.

7.       Did the Ethics Review Board approve the 5,000 won given to each participant, and why was compensation necessary in this survey?

Data analysis

1.       What means did you use to collate the data from the questionnaire into an “analysis-ready” spreadsheet?

2.       How did you manage missing values and univariate outliers?

3.       Did the authors try transformation for non-normally distributed variables?

4.       What were the assumptions of multiple linear regression met or not met by the data, and how were they diagnosed and fixed?

5.       It is good to know the mode of entry used for the regression model. Please list the predictors entered into the model such as demographic variables.

6.       Did you state the alpha level at which the decision of significance was set or did you supply confidence intervals?

 Results

1.       Another construct “job satisfaction” was first mentioned in the result section.

2.       The result presentation looks good, though tables are sometimes complex. Was Table 1 a mean QOL comparison between classes of the categorical variables? If yes, then the significant F-statistics should have the post hoc analysis reported in the text (the use of alphabets a,b<c is inadequate).

3.       The regression model (Table 4) needs to include sociodemographic variables seen in table 1.

4.       While dummy coding please take note of small and dominant classes. The fact that only 3(1.5%) of the people reported good subjective health status will affect both the regression and ANOVA models, please, consider binarizing such variables.

Discussion

The discussion was too shallow and out of context, and a greater part of the results was left out of the discussion. The introduction section can be reduced to allow discussion to gain more words if word count was the problem. Can the authors situate their findings in the Korean context? What are the strengths and weaknesses of the study, the clinical implications, and the policy implications? If you say the sleep, QOL, job satisfaction, and eating habit of nurses should be improved, what are the practical steps to this?

Conclusion

The conclusion was overstated, the data and analysis completed in this study cannot derive a causal effect as shown in the authors’ tone of writing. It doesn’t appear the influence of “physical, psychological and social factors” were exhausted to warrant a definite conclusion (see the definition in Lines 28 & 38).

Limitation

There should be a different section for limitations. Remove all the limitation discussions from the conclusion section.

References

The bibliography appears okay to me.

Author Response

I attached it as a file.

Reviewer 3 Report

Thank you for giving me the opportunity to read the paper. The title was interesting, and I enjoyed reading it. However, the significance of this study is not clear as there are few new parts in the results of this paper. Nonetheless, I think the paper can be improved. I would like to suggest some corrections. The specifics are as follows.

1. Please check the spacing in general. For example, in line 105, ‘ona5-point scale’ should be modified to ‘on a 5-point scale’. Please check the entire the paper.

Abstract.

2. Line 16-19: The meaning of the sentence is ambiguous. Isn't it clinical nurses who had a good subjective health status, not clinical nurses who had a subject health status? Is the meaning of the following sentence correct? “Multiple regression analysis showed that clinical nurses who had a good (or better) subjective health status, low perceived stress, and few sleep disturbances were more likely to have a higher quality of life.”

Introduction

3. Line 80: Please correct typos, “The analyses”.

Measures

4. Line 94: What is the version of the G*Power program?

5. Please write the author’s name of the tools consistently. Write ‘study by Park and Seoh’ on line 109 or write the full name of the author. Please write down the last name only or the full name of the authors of other tools, too.

Results

6. Line183 & Table1: I can't figure out how much '5 million won' is. Since it is not only read by Korean readers, please indicate in dollars.

7. Line 187: In the text, the special unit included the emergency room and the intensive care unit etc., but in the footnote of Table 1, the special unit is the emergency room and the intensive care unit only. Which one is correct?

8. In Table 1 subjective health status, the number of ‘good’ is only 3, but can it be said to be statistically significant?

9. Line 204-210: I’m not sure what you mean by item mean. Please explain why you added (an average of ~ per question).

10. Table 2: Is there a reason for adding Item mean in Table 2?

11. Line 225: Does univariate analysis mean ANOVA? If so, please use consistent terminology.

12. Line 227: you said eating behavior was a predictor variable, but in table 4, eating behavior was not significant. Please revise this sentence.

13. Line 236-237: Modify quality of life rather than subjective health status to be affected by perceived stress and sleep disturbances.

Discussion

14. Line 244-245: This appears to be a repetition of the above sentence. Please rewrite this sentence.

15. Line 246-247: I am not sure what this sentence means. Why does a higher level of health indicate a higher quality of life?

16. Line 272: What kind of research does previous studies mean in this sentence? Please add references.

17. Line 273-274: No. 25 of the reference is a paper with Kim as the first author and not related to sleep disturbance. Please check and correct.

18. Line 273-274: What part is the same as the literature of Lee et al.?

19. The quality of sleep may be low due to shift work, etc. Have you compared it with studies targeting nurses?

20. Eating behavior, which was also mentioned in the title, does not appear to be a factor affecting the quality of life, so it would be better to exclude it from the title. Also, please include in the discussion why eating behavior was not found to be significant in this study.

Conclusion

21. Line 282: - What does ‘the higher the positively perceived stress’ mean? According to the results of the study, perceived stress and quality of life are negative relations, and it seems correct to say that a lower perceived stress a higher quality of life.

22. Line 285: The wording of ‘increasing positively perceived stress’ also seems inappropriate. Please write again.

Author Response

I attached it as a file.

Reviewer 4 Report

Thanks to the  dear editor for the opportunity to review the article. The topic of the article is very interesting and useful. Although the article is well written, the following corrections are suggested to improve the quality of the article.

1- The formula for calculating the sample size should be mentioned in the method of working. Also, the criteria for inclusion the study and the criteria for exclusion the study should be mentioned

2- The discussion of the article is very poorly written and does not have a good coherence. Dear author, it is necessary to use more studies to compare and discuss the results of the study with a deeper search. The clinical application of the results of the study should be mentioned. The strengths and limitations of the study should be mentioned at the end of the discussion section

3- The conclusion should be based on the findings of the study. The conclusion should be corrected

4. Most of the references are not up-to-date, it is necessary to use more up-to-date studies

Author Response

I attached it as a file.

Round 2

Reviewer 3 Report

Thank you for your hard work. I think the authors reflected my opinion well in the paper. However, please double check the spacing in Line 259(QualityofLife).

Author Response

I checked again and corrected.
